# A Call for Preventing Suicide by Hanging from Ceiling Fans: An Interdisciplinary Research Agenda

**DOI:** 10.3390/ijerph16152708

**Published:** 2019-07-29

**Authors:** Kishan Kariippanon, Coralie J. Wilson, Timothy J. McCarthy, Kairi Kõlves

**Affiliations:** 1School of Health & Society, University of Wollongong, Wollongong NSW 2522, Australia; 2School of Graduate Medicine, University of Wollongong, Wollongong NSW 2522, Australia; 3School of Civil, Mining, & Environmental Engineering, University of Wollongong, Wollongong NSW 2522, Australia; 4Australian Institute for Suicide Research & Prevention, WHO Collaborating Centre for Research and Training in Suicide Prevention, Griffith University, Nathan QLD 4222, Australia

**Keywords:** means restriction, hanging, ceiling fan, engineering innovation, suicide

## Abstract

Hanging is a common method of suicide in several countries. Even as global suicide rates decrease, there is no evidence of suicides by hanging declining. There is limited research by type of hanging, and only a few papers present suicide by hanging from ceiling fans. Our paper proposes a research agenda that will: specify the size of the problem of hanging by ceiling fan (Stage 1: Surveillance), use standard engineering product development processes to modify ceiling fans for reducing their lethal capacity (Stage 2: Design Testing and Redevelopment), and examine the resulting beta- and release-build fans for safety and potential to reduce suicide in community samples (Stage 3: Evaluation).

## 1. Introduction

An estimated 800,000 suicide deaths occurred worldwide in 2016, and placed suicide in the top three causes of death among males and females aged 15–44 years, with an annual global age-standardised suicide rate of 10.7 per 100,000 population [1]. The highest rates per region were in Europe (15.4), followed by South East Asia (13.2) and the Western Pacific (10.2) [1]. While global suicide rates fluctuate over time, there appears to be a downward trend since 2000 [2], with exceptions in the USA [3] and Australia [4], where national suicide rates have continued to increase since 1999 and 2006, respectively. Suicide is a complex multifactorial phenomenon that makes it difficult to estimate the specific contribution of effective suicide prevention strategies or other changes on societal, community or individual rates. 

## 2. Hanging as a Suicide Method 

Despite the decline in global suicide rates, the number and proportion of suicides by hanging are rising. This highly lethal method has been reported as most prevalent in many countries worldwide, including many European countries [5], Australia [4,6], Canada [7], Asian countries, including India [8], South Korea [9] and Japan [10], and accounts for more than half of suicides in several of these countries. In the United Kingdom (UK), the proportion of suicides by hanging for females increased from 26.6% in 2001 to 42.8% in 2017, while the increase for men between 2001 and 2017 was from 44.5% to 60.6% [11]. The number of deaths by hanging has also increased in Australia by 33.7% from 1213 to 1829 (more than half of all suicides) between 2008 and 2017 [4].

Around the world, use of hanging is of particular concern in youth and indigenous communities who are at higher risk of suicide compared to nonindigenous populations [12]. In Australia, nearly 90% of Aboriginal and Torres Strait Islander people who die by suicide use hanging [13].

Globally, the relative increase in hanging proportion could be partly attributed to the reduction of the availability of other means, as several countries include means restriction in their national suicide prevention strategies [2,14,15]. It may also suggest the rare existence of a substitution effect in some countries [16]. For example, when means other than hanging were chosen, they were often less lethal (e.g., insufficient dose for suicide by overdose), resulting in fewer deaths. In contrast, hanging is often more physically accessible, easier to plan and cognitively more accessible, making it particularly difficult to restrict, except in controlled environments (e.g., prisons and hospitals) [17]. In this context, cognitive availability refers to the suicidal individual having knowledge and ability to implement suicide by hanging, alongside limited reasoning ability that would raise considerations such as painfulness [18]. In a qualitative study from the UK, Biddle and colleagues reported that people who nearly died by hanging suggested that their choice of means was because they perceived it to be a certain, quick, painless and ‘clean method’. This perception existed even though they also understood that they may not die, but may have instead a broken neck that would be painful and horrible [19].

### Hanging by Ceiling Fan 

Due to the sensitive nature and judicial requirements of reporting suicides, specific details about hanging suicides are not routinely available. Both suicides by hanging and suffocation are coded as X70 by ICD-10 [20], and are traditionally analysed together without stating the ligature point or further details. An extensive analysis of suicides by hanging in 16 US States between 2005 and 2014 found that ligature point was reported for only 30% of hanging suicides by corners or medical examiners [21]. Lack of details makes further consideration for target prevention activities, including innovations that could address suicide by different types of hanging challenging. 

The suicide mortality of hanging by ceiling fan has been rarely studied and reported. Limited evidence suggests that a household ceiling fan is a standard device used for hanging in dwellings of some countries in the Eastern Mediterranean Region (EMR) of WHO (i.e. Pakistan and Bahrain) [22]. In some countries, for example Bahrain, India and Pakistan, suicide by ceiling fan is reported to be one of the most common methods of hanging [22,23,24]. Some studies have reported that the proportion of hanging by ceiling fan in India is 10% and more common among females than males [24]; whereas, in the United States—where ligature point is recorded for only about one-third of hanging suicides—hanging by ceiling fan is reported to be relatively uncommon [21]. 

## 3. Research Agenda

The existing evidence points to hanging as a growing problem affecting a large proportion of the global population even though information about ligature points is currently limited. Since ceiling fans are commonly used in domestic and institutional settings in several countries as ligature points, we would like to make a call to action and propose a research agenda that aims to prevent suicides by hanging from ceiling fans. The research could start in Australia and expand to other countries. 

There could be several stages in a comprehensive research agenda: (1) Surveillance; (2) Design testing and redevelopment; (3) Evaluation. Surveillance data would identify the size of the problem of suicidal behaviour by ceiling fan, starting with retrospective data and moving towards prospective data. For example, in Australia, in addition to the National Coronial Information System [25]—which includes all reportable deaths—several states maintain suicide registers (e.g., Queensland Suicide Register) [6,13]. Data recorded in registries such as these would facilitate the analysis of profiles of those who die by suicide by ceiling fan compared to other means in different countries. Hospital records or self-harm registries (e.g., National Self-Harm Registry in Ireland) could also be used to examine attempted suicides, survival rates and factors contributing to survival [26]. Surveillance data would help to identify vulnerable groups and common processes that individuals have implemented to access and use ceiling fans for suicidal behaviour. Surveillance data would also be used to guide ceiling fan and building modifications as part of wider safety planning procedures. 

**Design testing and redevelopment** would first examine the mechanical elements of ceiling fans and their potential for hanging under different engineering conditions, as well as identify specific areas for engineering redevelopment to reduce the capacity of ceiling fans to act as reliable and lethal means [27]. Once specific areas for redevelopment are defined as concepts—for example, it could be helpful if current ceiling fan designs were modified to be non-weight bearing while upholding standards in building codes—the standard product development process for moving an engineering concept to market would be followed (i.e., systematic transition through the (i) concept; (ii) discovery; (iii) preliminary design; (iv) critical design; (v) build; (vi) test and (vii) beta build/release build product development phases). 

The **evaluation** would first examine the beta build ceiling fan in different environmental conditions (e.g., residential dwellings and prison settings) where surveillance data suggests the ceiling fan is often used for hanging. Once the beta build is deemed safe, the release build ceiling fan could be examined in a quasi-experimental study to evaluate the modified ceiling fan in residential and institutional settings, and in larger samples than used in beta testing. The main aim of an evaluation would be to examine the potential of the redeveloped fan for widespread community use. A potentially affordable modification to existing fans would be to build in a weak point in the connection of the fan support to the ceiling, designed to fail under the suddenly applied weight of an adult. An arrester cable or spring attached to the body of the fan and the ceiling could allow the fan to drop to a safe distance but not impact on the person. A long-term evaluation would assess the effect of modifications to the ceiling fans on the overall suicide rates and to identify any substitutions of alternative means.

While the problem we would like to specifically address through our agenda is hanging by ceiling fans, we believe that a dual strategy for suicide prevention, which also includes clinical services, has strong potential to reduce the total number of deaths by suicide around the world. As there are efforts to improve clinical services and global mental health policy [28], the primary focus of our agenda is on the modifications of ceiling fans.

## 4. Conclusions

Suicide by hanging is an increasing global challenge. We believe that focusing on means restriction, and specifically on reducing the lethal capacity of ceiling fans, alongside the ongoing improvement of access to clinical services, would contribute to the reduction of suicidal behaviour in Australia and around the world.

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
