# Peer review of "A Call for Preventing Suicide by Hanging from Ceiling Fans: An Interdisciplinary Research Agenda"

_ijerph, 2019, doi:10.3390/ijerph16152708_

Round 1

Reviewer 1 Report

This reserach agenda raises an interesting point of concern in suicide prevention, namely the rise of the use of ceiling fans for hanging. Given that means restriction is one of the most effective ways of reducing suicide, this proposal provides a novel opportunity for means restriction through the modification of the design of ceiling fans.

I have a few questions and points of clarification. Firstly, in section 2.1, for completeness, it may be worth noting why there is such a difference in the rate of hanging by ceiling fan between the noted countries. I assume this is largely due to the availability of firearms in the US, which are the major method of suicide (and as lethal or more lethal perhaps than hanging), whereas there is limited availability of these in the other countries. The % for Asia is also missing in this section.

I note that this agenda is somewhat 'Australia-centric' and I understand that most of the reserachers are from Australia; however, are you proposing that such work would best take place in Australia initially? If so, does starting this work in Australia have particular advantages (such as the type of data collection)? Alternatively, are you calling on researchers from other countries to join in this reserach agenda? What is the particular purpose of promoting this research agenda? If this a 'call to action', then a clearer message regarding what you are asking of others might be more effective in motivating action.

In regard to the evaluation, would it also be useful to state that a longer term evaluation agenda would be to assess the effect of modifications to ceiling fans of the overall suicide rate and to identify any substitutions of alternative means?

You also mention access to clinical services in passing. It may be useful to flesh out this component or purely state that while you believe this work needs to take place parallel to the work on alternative ceiling fan design, that the focus of your work is on ceiling fans because the other work is taking place elsewhere (or something similar).

I think the partnership of researchers from health and engineering in this research agenda is a great example for other suicide prevention researchers.

Reviewer 2 Report

This is a very interesting short report on ceiling fans as a potential target for hanging means restriction for suicide prevention - and is important because hanging is the most prominent means of suicide death in many countries. It is well written and structured, and provides a nice concise overview of the issue and directions for future work in this area. There are, however, some issues that require clarification. These are: 

In Section 2.1. (page 2) the authors present some statistics around the prevalence of ceiling fan deaths  - it is unclear whether these are reporting proportions of ceiling fans within hanging deaths only, or whether these are a proportion of all suicide deaths. This needs to be made clearer to build a stronger rationale of why it is important to be looking at ceiling fan deaths as a target of prevention efforts (i.e., the scope/size of the problem is unclear to me at the moment - needs to be more explicit).     

If possible, it would also be good in section 2.1. to present data on where these ceiling fan deaths are happening (home, away? if the latter, where?). This would also have implications for means restriction as this will be harder to monitor in homes than in clinical or justice settings. If such data is not available, I think it's important to highlight how little is known about these types of deaths. 

In section 3, page 2: The first sentence suggests that ceiling fan deaths are a global phenomenon - however, in the previous section, the authors have only presented evidence to indicate this is a problem in Eastern countries. As such, I think this opening sentence should be amended to reflect that, based on available data, this appears to be a phenomenon in Eastern countries (rather than a global problem, which is a potential overstatement).

In Section 3, page 2: This paper is framing ceiling fan deaths as a 'global' problem, but then in the potential solution to the problem - surveillance - this is only discussed in the context of Australia. Other countries have suicide registries, it would make more sense in this paper to discuss the use of such registries to start to monitor the problem and establish a global understanding of the extent of the problem + the specifics of it, to identify opportunities for targeted means restriction strategies. Have the authors given any consideration as to how surveillance data may translate into action (i.e., how will knowledge transfer be facilitated into action).

If ceiling fans were made 'safer' what evidence is there to suggest that these individuals would not simply transfer to a different, more reliable method of hanging (i.e., using door frames/doors etc). This is not substitution per se, as individuals would still ultimately be choosing hanging as the method of death. Given this is a potential risk, are there broader implications to be addressed here around safety planning in the home?   

As a very minor point, on page 2, line 51, I am unsure of what 'cognitively more accessible' means. Is it possible to clarify this in the paper? 

Round 2

Reviewer 2 Report

I am happy with the revisions and this paper will make an interesting, new contribution to the suicide prevention literature. No further edits required.